# Relationship between Vitamin D Level and Lipid Profile in Non-Obese Children

**DOI:** 10.3390/metabo9070125

**Published:** 2019-06-30

**Authors:** Mi Ra Kim, Su Jin Jeong

**Affiliations:** Department of Pediatrics, CHA Bundang Medical Center, CHA University School of Medicine, Seongnam 13496, Korea

**Keywords:** vitamin D deficiency, dyslipidemia, non-obese children

## Abstract

Vitamin D deficiency is associated with not only cardiovascular disease itself but also cardiovascular risk factors, including obesity, hypertension, diabetes, hyperglycemia, and dyslipidemia. This study aimed to investigate the relationship between vitamin D level and lipid profile in non-obese children. A total of 243 non-obese healthy volunteers, aged 9–18 years, were enrolled from March to May 2017. Their height and weight were measured, and body mass index was calculated. Subjects underwent blood tests, including measurements of vitamin D (25(OH)D) level and lipid panels, and were divided into either the vitamin D-deficient group (<20 ng/mL) or normal group. The student’s *t*-test and a simple linear regression analysis were used to estimate the association between vitamin D level and lipid profile. Overall, 69.5% of non-obese children (*n* = 169) had a 25(OH)D level of less than 20 ng/mL. The vitamin D-deficient group showed higher triglyceride (TG) level and TG/high-density lipoprotein cholesterol (HDL-C) ratio than the normal group (TG level: 90.27 vs. 74.74 mmol/L, *p* = 0.003; TG/HDL-C ratio: 1.753 vs. 1.358, *p* = 0.003). Vitamin D level seems to affect the lipid profile, even in non-obese children, and a low vitamin D level may progress to dyslipidemia or obesity in non-obese children.

## 1. Introduction

Vitamin D, a secosteroid, plays a pivotal role in the protection against numerous diseases, including cardiovascular diseases. More than 80% of metabolic vitamin D is derived from sunlight, whereas the remaining percentage is acquired via dietary supplementation. There has been an augmented interest in vitamin D within the medical community, particularly in the relationship between vitamin D deficiency and various systemic disorders. Previous studies have reported a prevalence rate of 30–50% for vitamin D deficiency among adults [1,2]. Furthermore, 32% of the general pediatric population and 74% of obese children reportedly have vitamin D deficiency [3].

Increased body mass index (BMI) has been consistently shown to be associated with lower vitamin D concentrations [4,5,6], which can be elucidated by the fact that obese children exercise less than their peers and are hence less exposed to sunlight, with limited cutaneous vitamin D synthesis and lower bioavailability of fat-soluble vitamin D sequestered in the fat compartment [7]. Moreover, obesity itself leads to several metabolic diseases, including hypertension, diabetes, hyperglycemia, and dyslipidemia, and several studies on obese adults have indicated a correlation between serum vitamin D level and lipid profile.

Vitamin D deficiency is associated with not only cardiovascular disease itself but also cardiovascular risk factors [8,9]. Low vitamin D levels could result in dyslipidemia, and lipid abnormalities—that is, an increase in triglyceride (TG), total cholesterol (TC), and low-density lipoprotein cholesterol (LDL-C) levels and a decrease in high-density lipoprotein cholesterol (HDL-C) level—have been identified as important risk factors for atherosclerosis and cardiovascular disease in adulthood [10,11]. 

Adult diseases and their risk factors have been well documented to originate in early life [12], considering the high prevalence of vitamin D deficiency in children and adolescents [13] and the growing prevalence of cardiometabolic risk factors in the pediatric age group [14,15]. However, limited information on the association between vitamin D levels and cardiometabolic risk factors in the pediatric population is available [16,17]. Evaluating the relationship between vitamin D and cardiometabolic risk factors in early life can aid in the acquisition of a better understanding about underlying mechanisms and in the performance of action-oriented interventions for primordial and primary prevention of several chronic diseases. Longitudinal studies have shown that dyslipidemia during childhood often persists into adulthood and is associated with cardiometabolic diseases; consequently, there has been an increasing focus on lipid screening and intervention [18,19]. Therefore, early identification of dyslipidemia is crucial in halting the progression of cardiovascular disease and maintaining health in the long term.

Some studies investigating the association between vitamin D levels and dyslipidemia in obese children have suggested that vitamin D deficiency might be linked to metabolic syndrome [20]. Nevertheless, there is a lack of studies on the relationship between vitamin D levels and lipid panels in non-obese children. Hence, the present study aimed to investigate the relationship between serum vitamin D (25-hydroxyvitamin D [25(OH)D]) level and lipid profile in non-obese children.

## 2. Experimental Section

### 2.1. Study Population

The study was performed from March to May 2017. A total of 243 non-obese healthy volunteers (117 boys and 126 girls) aged 9–18 years (mean age, 11.26 [SD, 1.91] years) were recruited from one elementary school and two middle schools in Seongnam-si, Gyeonggi-do, Korea. Subjects did not have any history of malignancies, hypertension, diabetes, severe liver and kidney diseases, and other conditions that could affect vitamin D levels (e.g., hyperparathyroidism, hyperthyroidism). Children aged <9 years who exhibited difficulty in observing the overnight fast required for the blood test and had insufficient capacity for understanding the study procedure were excluded from the analysis.

When designing this study, we were inspired by the investigation done by Hirschler V. et al. [21]. Therefore, we calculated the sample size based on their study. To obtain the power of 90%, i.e., *p* < 0.05, we needed more than 58 subjects in each group. In this study, we divided the subjects into two groups. The vitamin D-deficient group had 169 subjects, and the vitamin D normal group had 74 subjects.

### 2.2. Anthropometric and Laboratory Assessment

Anthropometric measurements were conducted with the children wearing light clothing and no shoes. One well-trained examiner measured height and weight to the nearest 0.5 cm and 0.5 kg, respectively, using an automated height–weight scale (DS-102; Dong Sahn Jenix, Seoul, Korea). Non-obese children were defined as those with BMI < 23 kg/m^2^. 

BMI is the most widely used measure of obesity is with the World Health Organization cut-offs defined as 18.5–24.9 kg/m^2^ for normal individuals. However, these values are largely based on the morbidity and mortality data from the white Caucasian populations and may not be applicable across all ethnic groups. Moreover, the proposed cut-offs defining overweight and obesity are not appropriate for Asian men because they are at a risk of developing obesity-related co-morbidities at a lower BMI. Misra et al. have reported consensus regarding BMI cut-offs normal up to 22.9 kg/m^2^ based on the seven studies reviewed [22,23].

Blood samples were drawn from the antecubital vein after 8 h of overnight fasting. Fasting plasma lipid concentrations, including TG and HDL-C levels, were analyzed using an autoanalyzer (Cobas 8000 modular analyzer series; Roche Diagnostics, Basel, Switzerland), whereas 25(OH)D levels were measured using chemiluminescent immunoassay (Atellica Solution; Siemens Healthineers, Erlangen, Germany).

### 2.3. Statistical Analysis

The statistical analysis was performed using SPSS versions 24 and 25 (IBM Corp., Armonk, NY, USA). Student’s *t*-test and simple linear regression analysis were used to estimate the association between the serum vitamin D level and the lipid profile. Statistical significance was set at *p* < 0.05.

### 2.4. Ethical Considerations

Written informed consent was obtained from the parents of all subjects enrolled in the study. The study was approved by the CHA University Institutional Review Board (CHAIRB No.2015-196).

## 3. Results

### 3.1. Subject Characteristics

Characteristics of subjects who participated in this study are summarized in Table 1. A total of 243 subjects were identified as non-obese children with BMI < 23 kg/m^2^ (mean BMI, 18.82 [SD, 2.38]). For these 243 non-obese children, the mean 25(OH)D, TC, LDL-C, HDL-C, and TG levels were 17.27 ng/mL (4.4–43.0 ng/mL), 169.16 mg/dL (66.2–286.0 mg/dL), 94.93 mg/dL (23.0–177.0 mg/dL), 57.38 mg/dL (18.8–114.5 mg/dL), and 85.54 mg/dL (28.0–346.0 mg/dL), respectively. Furthermore, the mean TG/HDL-C ratio was 1.63 (0.32–9.91). No difference in age or sex was observed.

### 3.2. Association between Vitamin D Level and Lipid Profile

Vitamin D level was associated with TC (β coefficient = −0.126, *p* = 0.658), LDL-C (β coefficient = −0.204, *p* = 0.376), and HDL-C levels (β coefficient = 0.207, *p* = 0.089), albeit without statistical significance. Nevertheless, the vitamin D level was significantly inversely associated with TG level (β coefficient = −1.506, *p* < 0.001) and TG/HDL-C ratio (β coefficient = −0.041, *p* < 0.001) With respect to TG level and TG/HDL-C ratio, vitamin D level was significantly correlated to lipid profile (Figure 1).

### 3.3. Comparison of Vitamin D Levels Between Two Groups

Subjects were divided into two groups—namely, the vitamin D-deficient group and the normal group. Overall, 69.5% of non-obese children (*n* = 169) had a 25(OH)D level of less than 20 ng/mL. The results for cholesterol (TC, HDL-C, and LDL-C) levels did not reach statistical significance (Table 2). However, the vitamin D-deficient group had higher TG levels and a higher TG/HDL-C ratio than the normal group (TG level: 90.27 vs. 74.74 mmol/L, *p* = 0.003; TG/HDL-C ratio: 1.753 vs. 1.358, *p* = 0.003). With respect to TG level and TG/HDL-C ratio, vitamin D level was significantly related to lipid profile (Figure 2).

## 4. Discussion

The results of this study indicated a significant association between serum 25(OH)D level and lipid profile, particularly TG level and TG/HDL-C ratio, in children and adolescents without obesity. 

Although 1,25-dihydroxyvitamin D is recognized as the active form of vitamin D, 25(OH)D, the primary circulating form of vitamin D, it is considered the best indicator of serum vitamin D level and a sensitive measure of serum vitamin D status because of its long half-life. There exists no international definition for optimal vitamin D status in children. We divided subjects into two groups: Vitamin D-deficient group (25(OH)D <20 ng/mL) and the normal group (25(OH)D ≥20 ng/mL). In our study, the mean 25(OH)D level was 17.27 ng/mL, and there were more vitamin D-deficient children, which is similar to the result of the 2008 Korea National Health and Nutrition Examination Survey [24]. 

Dyslipidemia, a cardiovascular risk factor, is characterized by elevated TG and LDL-C levels and reduced HDL-C levels. Small dense LDL (sdLDL) tends to rapidly deposit on the arterial wall, reducing LDL-C clearance. Hence, sdLDL is correlated to atherosclerosis and coronary artery disease. The atherogenic index of plasma (AIP), expressed as log[TG/HDL-C], could be an excellent predictor of sdLDL level. In the study of Wang et al., AIP > 0.15 was regarded as abnormal [25]. Therefore, the TG/HDL-C ratio can be useful for predicting cardiovascular risk in the future. In our study, 113 (46.5%) out of 243 subjects had an abnormal TG/HDL-C ratio and thus had cardiovascular risk. Furthermore, 83 (49.1%) out of 169 vitamin D-deficient children had cardiovascular risk.

Previous studies have reported similar results for the adult population. Wang et al. investigated the effects of serum vitamin D status on lipids in Chinese adults and showed that serum 25(OH)D level was closely associated with lipids and AIP. In addition, Chaudhuri et al. reported that 25(OH)D deficiency was independently associated with dyslipidemia in Indian subjects [26]. Numerous studies have confirmed the association between vitamin D level and lipid panel in children and adolescent with obesity. In the study of Lee et al., the mean 25(OH)D level was lower in children deemed obese. As the vitamin D level increased, TG and HDL-C levels became lower and higher, respectively [27]. Rusconi et al. reported the association between low 25(OH)D level and unfavorable lipid patterns in a pediatric obese population [20]. In our study, we also observed a relationship between vitamin D level and dyslipidemia even in non-obese children and adolescents.

The functions of vitamin D are linked to lipid values. First, vitamin D regulates calcium metabolism and increases intestinal calcium absorption, thereby reducing intestinal fatty acid absorption [25]. Therefore, a reduction in intestinal fat absorption can lower the cholesterol level. Additionally, increasing the calcium concentration promotes the conversion of cholesterol into bile acids in the liver, resulting in reduced cholesterol level [28].

Second, high vitamin D level inhibits the parathyroid hormone (PTH). When the vitamin D level is not high, vitamin D may not inhibit PTH [29,30]. Increased PTH level enhances lipogenesis, promoting calcium influx into the adipocytes. Furthermore, a high PTH level decreases lipolytic activity, resulting in a high TG level. Therefore, in the presence of high vitamin D level, a low PTH level can reduce the TG level by increasing lipolytic activity and peripheral removal. In addition, a high PTH level increases bone turnover and induces calcium release from the bone. Increasing the calcium concentration can affect the cholesterol level, as explained above.

Third, vitamin D can affect lipoprotein metabolism and reduce TG synthesis and secretion in the liver, increasing very-low-density lipoprotein (VLDL-C) receptor expression. Consequently, a high vitamin D level induces a decrease in TG and VLDL-C levels and an increase in HDL-C level.

Our study showed that vitamin D could influence the lipid profile, even in non-obese children. Vitamin D-deficient children had a higher TG level and TG/HDL-C ratio and, therefore, they may progress to dyslipidemia or obesity. Dyslipidemia during childhood persists into adulthood; hence, maintaining the vitamin D level within the normal range seems important, even for non-obese children.

Lee et al. reported that hypertriglyceridemia and high TG/HDL-C ratio could increase the risk of nonalcoholic fatty liver disease (NAFLD) [31]. Hence, it could be supposed that low vitamin D level could induce the progression to metabolic diseases, such as NAFLD. Therefore, a normal 25(OH)D level may be a marker of better lipid profile and reduced risk of adult diseases in non-obese children. 

Our study limitation was that we did not consider diet differences and lifestyle habits, which may have influenced the relationship between vitamin D level and lipid profile. Future studies analyzing the association between vitamin D levels and lipid profile are warranted. Furthermore, the sample size of this investigation was small even though it was enough to conduct the study. Therefore, a large cohort study would be needed to support these results.

Whether dietary vitamin D supplementation plays a therapeutic or preventive role remains poorly explored. Data from epidemiologic studies [32,33] do not support the finding that 25(OH)D supplementation could beneficially improve the lipid profile, thus leaving the role of 25(OH)D uncertain [34]. A longitudinal follow-up study is required in order to verify whether vitamin D deficiency could lead to obesity or complications such as cardiovascular disease. Further studies on the potential positive effects of vitamin D supplementation on the lipid profile are warranted.

## 5. Conclusions

The results of this study partially confirmed the negative association between 25(OH)D level and lipid profile, including TG level and TG/HDL-C ratio. Furthermore, our findings indicated the relationship between vitamin D levels and lipid profile, even in non-obese children. 

Owing to a growing obese pediatric population, there has been an increasing interest in the prevention of adult diseases. Nonetheless, the prevention of adult diseases and management of risk factors for cardiovascular disease tend to be unnecessary in non-obese children. As suggested by our study findings, the maintenance of vitamin D level within an appropriate range is vital for the prevention of adult diseases, even in non-obese children and adolescents.

## Figures and Tables

**Figure 1 metabolites-09-00125-f001:**
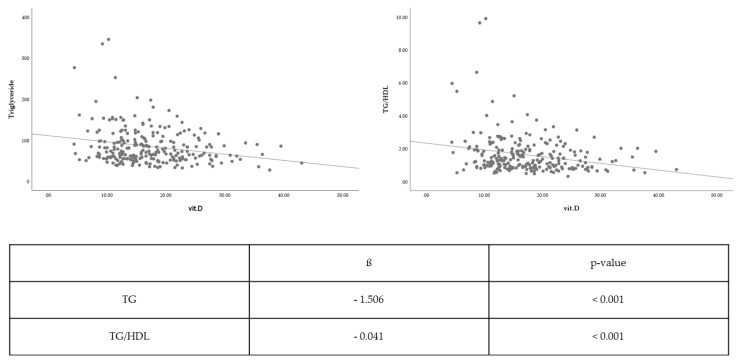
Association between the vitamin D levels and lipid profiles. TG/HDL: Triglyceride to high-density lipoprotein, vit.: Vitamin.

**Figure 2 metabolites-09-00125-f002:**
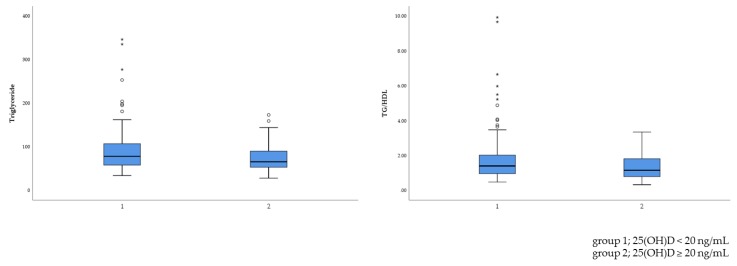
Comparison of the lipid profiles at different vitamin D levels. TG/HDL: Triglyceride to high-density lipoprotein.

**Table 1 metabolites-09-00125-t001:** Subject characteristics.

Variables	Subjects (*n* = 243)
	Mean (Range, SD)
Age (years)	11.26 (9–18, 1.91)
Anthropometric profile	
Height (cm)	146.05 (120.8–182.1, 10.5)
Weight (kg)	40.47 (22–68.8, 8.51)
BMI (kg/m^2^)	18.82 (11.93–22.94, 2.38)
Laboratory profile	
25(OH)D (ng/mL)	17.27 (4.4–43, 6.89)
TC (mg/dL)	169.16 (66.2–286, 30.43)
LDL-C (mg/dL)	94.93 (23–177, 24.65)
HDL-C (mg/dL)	57.38 (18.8–114.5, 13.01)
TG (mg/dL)	85.54 (28–346, 45.13)
TG/HDL-C	1.63 (0.32–9.91, 1.2)

BMI: Body mass index, TC: Total cholesterol, LDL-C: Low-density lipoprotein cholesterol, HDL-C: High-density lipoprotein cholesterol, TG: Triglyceride, TG/HDL-C: Triglyceride to high-density lipoprotein cholesterol.

**Table 2 metabolites-09-00125-t002:** Comparison of the characteristics at different vitamin D levels.

Variables	25(OH)D < 20ng/mL (*n* = 169)	25(OH)D Normal (*n* = 74)	*p*-Value
	Mean (Range, SD)	Mean (Range, SD)	
Age (years)	11.36 (9–18, 2.07)	11.04 (9–16, 1.48)	-
Anthropometric profile	-	-	-
Height (cm)	146.84 (121.1–178, 10.59)	144.26 (120.8–182.1, 10.12)	-
Weight (kg)	40.78 (22–65, 8.91)	39.76 (25.8–68.8, 7.54)	-
BMI (kg/m^2^)	18.73 (11.93–22.94, 2.43)	19.01 (12.49–22.62, 2.27)	-
Laboratory profile	-	-	-
25(OH)D (ng/mL)	13.58 (4.4–19.9, 3.62)	25.68 (20–43.03, 4.93)	-
TC (mg/dL)	168.97 (66.2–286, 30.39)	169.59 (77–235, 30.74)	0.883
LDL-C (mg/dL)	95.04 (23–177, 24.71)	94.66 (31–158, 24.68)	0.911
HDL-C (mg/dL)	56.64 (18.8–98.4, 12.53)	59.09 (32.5–114.5, 14)	0.178
TG (mg/dL)	90.27 (34–346, 49.4)	74.74 (28–173, 31.05)	0.003
TG/HDL-C	1.75 (0.48–9.91, 1.35)	1.36 (0.32–3.34, 0.7)	0.003

BMI: Body mass index, TC: Total cholesterol, LDL-C: Low-density lipoprotein cholesterol, HDL-C: High-density lipoprotein cholesterol, TG: Triglyceride, TG/HDL-C: Triglyceride to high-density lipoprotein cholesterol.

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
