# Peer review of "Relationship between Vitamin D Level and Lipid Profile in Non-Obese Children"

_metabolites, 2019, doi:10.3390/metabo9070125_

Round 1
Reviewer 1 Report
This manuscript describes the association between serum 25(OH)D level and lipid profile in non-obese children. This topic is of relevant interest in order to prevent adult diseases (e.g. cardiovascular disease) considering the important protection role of this nutrient.
Relevant methodological pitfalls and inaccuracies are present and should necessarily be clarified:
Introduction: lines 40-45; 53-55; 58-59: please add literature concerning these sentences.
Study population: sample size calculation is lacking. Please add this important information.
Line 77: “Non-obese children were defined as those with BMI <23 kg/m2.”: what is the reference for this diagnostic criteria? Maybe it is better to refer to WHO diagnostic criteria (e.g. Cut-offs for children 5-19 years: Obesity, BMI-for-age: >+2SD; Overweight, BMI-for-age: >+1SD). Please reconsider this aspect for the analysis.
Author Response
Point 1: In Introduction, lines 40-45, 53-55, and 58-59, please add literature concerning these sentences.
Response 1:
Lines 40-45: Vitamin D deficiency is associated with not only cardiovascular disease itself but also cardiovascular risk factors [8,9]. Low vitamin D levels could result in dyslipidemia, and lipid abnormalities—that is, an increase in triglyceride (TG), total cholesterol (TC), and low-density lipoprotein cholesterol (LDL-C) levels and a decrease in high-density lipoprotein cholesterol (HDL-C) level—have been identified as important risk factors for atherosclerosis and cardiovascular disease in adulthood [10,11].
8. Dobnig, H.; Pilz, S.; Scharnagl, H.; Renner, W.; Seelhorst, U.; Wellnitz, B.; Kinkeldei, J; Boehm, B.O.; Weihrauch, G.; Maerz, W. Independent association of low serum 25-hydroxyvitamin d and 1,25-dihydroxyvitamin d levels with all-cause and cardiovascular mortality. Arch Intern Med 2008, 168, 1340–1349.
9. Martini, L.A.; Wood, R.J. Vitamin D status and the metabolic syndrome. Nutr Rev 2006, 64, 479–486, doi: 10.1111/j.1753-4887.2006.tb00180.x
10. Polkowska, A.; Głowińska-Olszewska, B.; Tobiaszewska, M.; Bossowski, A. Risk factors for cardiovascular disease in children with type 1 diabetes in 2000-2010 in Podlasie Province. Pediatr Endocrinol Diabetes Metab 2014, 20, 47–54, doi:10.18544/PEDM-20.02.0002.
11. Potenza, M.V.; Mechanick, J.I. The metabolic syndrome: definition, global impact, and pathophysiology. Nutr Clin Pract 2009, 24, 560–577.
Lines 53-55: Longitudinal studies have shown that dyslipidemia during childhood often persists into adulthood and is associated with cardiometabolic diseases; consequently, there has been an increasing focus on lipid screening and intervention [18,19].
18. Hirschler, V.; Maccallini, G.; Molinari, C.; Aranda, C.; San Antonio de los Cobres Study, G. Low vitamin D concentrations among indigenous Argentinean children living at high altitudes. Pediatr Diabetes 2013, 14, 203–210, doi:10.1111/pedi.12004.
19. Mark, S.; Gray-Donald, K.; Delvin, E.E.; O'Loughlin, J.; Paradis, G.; Levy, E.; Lambert, M. Low vitamin D status in a representative sample of youth from Quebec, Canada. Clin Chem 2008, 54, 1283–1289, doi:10.1373/clinchem.2008.104158.
Lines 58-59: Some studies investigating the association between vitamin D levels and dyslipidemia in obese children have suggested that vitamin D deficiency might be linked to metabolic syndrome [20].
20. Rusconi, R.E.; De Cosmi, V.; Gianluca, G.; Giavoli, C.; Agostoni, C. Vitamin D insufficiency in obese children and relation with lipid profile. Int J Food Sci Nutr 2015, 66, 132–134, doi:10.3109/09637486.2014.959902.
Point 2: In Study population, sample size calculation is lacking. Please add this important information.
Response 2: When designing this study, we were inspired by the investigation done by Hirschler V. et al. [21]. Therefore, we calculated the sample size based on their study. To obtain the power of 90%, i.e., p < 0.05, we needed more than 58 subjects in each group. In this study, we divided the subjects into two groups. The vitamin D-deficient group had 169 subjects, and vitamin D normal group had 74 subjects.
21. Hirschler, V.; Maccallini, G.; Sanchez, M.S.; Castano, L.; Molinari, C. Improvement in high-density lipoprotein cholesterol levels in argentine Indian school children after vitamin D supplementation. Horm Res Paediatr 2013, 80, 335–342, doi:10.1159/000355511.
Point 3: On Line 77, “Non-obese children were defined as those with BMI <23 kg/m2.”, what is the reference for this diagnostic criteria? Maybe it is better to refer to WHO diagnostic criteria (e.g. Cut-offs for children 5-19 years: Obesity, BMI-for-age: >+2SD; Overweight, BMI-for-age: >+1SD). Please reconsider this aspect for the analysis.
Response 3: BMI is the most widely used measure of obesity is with the World Health Organization cut-offs defined as 18.5-24.9 kg/m2 for normal individuals. However, these values are largely based on the morbidity and mortality data from the white Caucasian populations and may not be applicable across all ethnic groups. Moreover, the proposed cut-offs defining overweight and obesity are not appropriate for Asian men because they are at a risk of developing obesity-related co-morbidities at a lower BMI. Misra et al. have reported consensus regarding BMI cut-offs normal up to 22.9 kg/m2 based on the 7 studies reviewed [22,23].
22. Misra, A.; Chowbey, P.; Makkar, B.M.; Vikram, N.K.; Wasir, J.S.; Chadha, D.; Joshi, S.R.; Sadikot, S.; Gupta, R.; Gulati, S.; et al. Consensus statement for diagnosis of obesity, abdominal obesity and the metabolic syndrome for Asian Indians and recommendations for physical activity, medical and surgical management. J Assoc Physicians India. 2009, 57, 163–170.
23. Banerji, M.A.; Faridi, N.; Atluri, R.; Chaiken, R.L.; Lebovitz, H.E. Body composition, visceral fat, leptin, and insulin resistance in Asian Indian men. 1999, 84, 137–144.

Reviewer 2 Report
The manuscript entitled <Relationship between vitamin D level and lipid profile in non-obese children> is well explained, the authors have presented all their results in a correct and scientific rigurous manner. Nonetheless, the reduced sample size and the few experiments performed does not complies with the criteria to consider this manuscript as a full research article. Nontheless it can be submitted as short communication or increased in the results for a new evaluation.
A few considerations should be adressed by the authors:
Besides the clinical history, did the authors considered socioeconomic equality, health habits, and diet differences in the elaboration of experimental groups?
Do the authors considered adequate the statistical method apply to these data? Is a student's t test the correct test to treat data, enough to consider the results as statistically significant?
Correlations in section 3.2 deserve a deeper explanation.
I believe more parameters are necessary to support the conclussions, the authors should have in consideration the diet, health habits and other particular conditions to not let conclussions remains incomplete.
Author Response
Point 1: Besides the clinical history, did the authors considered socioeconomic equality, health habits, and diet differences in the elaboration of experimental groups?
Response 1: We recruited subjects from three schools in Seongnam-si, Gyeonggi-do, Korea. So children in our study group lived and went to school in same area. In Seongnam-si, the socioeconomic status is similar each other. But, about health habits and diet differences, we would include the limitations, “Our study limitation was that we did not consider diet differences and health habits, which may have influenced the relationship between vitamin D level and lipid profile. Future studies analyzing the association between vitamin D levels and lipid profile are warranted.”.
Point 2: Do the authors considered adequate the statistical method apply to these data? Is a student's t test the correct test to treat data, enough to consider the results as statistically significant?
Response 2: We used a student’s t test and simple linear regression analysis to estimate the association between serum vitamin D level and lipid profile. Especially, a student’s t test was used for comparing two groups; vitamin D deficient and normal. Because each group had more than 30 subjects, we did not need a normality anaylsis. So a student’s t test was the adequate statistical analysis and there had statistical significance on p < 0.05.
Point 3: Correlations in section 3.2 deserve a deeper explanation.
Response 3: We would add the sentence “ With respect to TG level and TG/HDL-C ratio, vitamin D level was significantly correlated to lipid profile. “ emphasizing the results as section 3.3
Point 4: I believe more parameters are necessary to support the conclussions, the authors should have in consideration the diet, health habits and other particular conditions to not let conclussions remains incomplete.
Response 4: Diet and health habits which result in obesity and dyslipidemia make vitamin D low. That is due to reduced the absorption of vitamin D. Our aim is that although children are not in obesity status, vitamin D level could have influence to lipid levels. So in the future, vitamin D deficient children could progress to be obese or have dyslipidemia.
Nevertheless, we agree with your opinion. So about diet differences and health habits, we would include the limitations, “Our study limitation was that we did not consider diet differences and health habits, which may have influenced the relationship between vitamin D level and lipid profile. Future studies analyzing the association between vitamin D levels and lipid profile are warranted.”.
But, about other particular conditions, we think that at young age, to assess other metabolic risk is difficult and for that, longitudinal cohort studies would be needed. We explained that on the end of discussion, “A longitudinal follow-up study is required in order to verify whether vitamin D deficiency would lead to obesity or complications such as cardiovascular disease.”.

Round 2
Reviewer 1 Report
The current manuscript could be published: all comments have been clarified
Author Response
Point 1:
The current manuscript could be published: all comments have been clarified.
Response 1:
Thanks for your review.

Reviewer 2 Report
The authors of the manuscript entitled <Relationship between vitamin D level and lipid profile in non-obese children> have considered the recommendations of this reviewer, at least partially. Although no relevant changes have been performed in the manuscript, in regard of lipemia and vitamin D relationship with other nutritional parameters, not even in the bibliographic research.
It is this reviewer consideration that even though the paper is well explained and results are in some manner of interest for nutritional studies (if complete), the authors should emphazise the fact that this is a preliminary study in a short scheme of communication, and indicate the future remarks if they are plannig to publish more results or continue the study as with this information it is not able to make any formal statement that brings novel and relevant information for other researchers.
I believe is worth for publication, but still firm in the conviction that not as full research article, as in the guidelines of this Journal it can be read that < short Communications of preliminary, but significant, results will be considered>, which is clearly this manuscript's case. Under consideration of the Editor of this journal.
Author Response
Point 1:
The authors have considered the recommendations of this reviewer, at least partially. Although no relevant changes have been performed in regard of lipemia and vitamin D relationship with other nutritional parameters, not even in the bibliographic research.
It is this reviewer consideration that even though the paper is well explained and results are in some manner of interest for nutritional studies (if complete), the authors should emphazise the fact that this is a preliminary study in a short scheme of communication.
I believe is worth for publication, but still firm in the conviction that not as full research article, as in the guidelines of this Journal it can be read that < short Communications of preliminary, but significant, results will be considered>.
Response 1:
We agree with your opinion and so agree with changing this manuscript to other article type: communication.
You mentioned, in last review, we needed to consider other parameters such as health habits and diet differences. So we added the limitations as in the following:
“Our study limitation was that we did not consider diet differences and health habits, which may have influenced the relationship between vitamin D level and lipid profile. Future studies analyzing the association between vitamin D levels and lipid profile are warranted.”.
As for other metabolic parameters, we think that to consider metabolic risk factors is difficult at young age and rather, longitudinal studies would be needed. On the end of discussion, we explained that, “A longitudinal follow-up study is required in order to verify whether vitamin D deficiency would lead to obesity or complications such as cardiovascular disease.”.
About your point that this is a preliminary study, we would add the mention, “The sample size of this investigation is small even though it is enough to study. So large cohort study would be needed to support this results.” Also, the last paragraph, “Whether dietary vitamin D supplementation plays a therapeutic or preventive role remains poorly explored. Further studies on the potential positive effects of vitamin D supplementation on the lipid profile are warranted.”, we think that might deal with your point.
Round 3
Reviewer 2 Report
In this version of the manuscript, the authors have included all the relevant information to state the kind of publication. I appreciate their effort in this regard, which in this particular case, after revison and changes in the type of submission, i recommend its acceptance in the present form.